# Salt-Tolerant Plants as Sources of Antiparasitic Agents for Human Use: A Comprehensive Review

**DOI:** 10.3390/md21020066

**Published:** 2023-01-19

**Authors:** Maria João Rodrigues, Catarina Guerreiro Pereira, Marta Oliveira, Gökhan Zengin, Luísa Custódio

**Affiliations:** 1Centre of Marine Sciences (CCMAR), Universidade do Algarve, Faculdade de Ciências e Tecnologia, Ed. 7, Campus de Gambelas, 8005-139 Faro, Portugal; 2Department of Biology, Science Faculty, Selcuk University, 42250 Konya, Turkey

**Keywords:** ethnomedicine, halophytes, helminthiases, parasitosis, salinization

## Abstract

Parasitic diseases, especially those caused by protozoans and helminths, such as malaria, trypanosomiasis, leishmaniasis, Chagas disease, schistosomiasis, onchocerciasis, and lymphatic filariasis, are the cause of millions of morbidities and deaths every year, mainly in tropical regions. Nature has always provided valuable antiparasitic agents, and efforts targeting the identification of antiparasitic drugs from plants have mainly focused on glycophytes. However, salt-tolerant plants (halophytes) have lately attracted the interest of the scientific community due to their medicinal assets, which include antiparasitic properties. This review paper gathers the most relevant information on antiparasitic properties of halophyte plants, targeting human uses. It includes an introduction section containing a summary of some of the most pertinent characteristics of halophytes, followed by information regarding the ethnomedicinal uses of several species towards human parasitic diseases. Then, information is provided related to the antiprotozoal and anthelmintic properties of halophytes, determined by in vitro and in vivo methods, and with the bioactive metabolites that may be related to such properties. Finally, a conclusion section is presented, addressing perspectives for the sustainable exploitation of selected species.

## 1. Introduction

Human parasitic diseases are predominantly linked to tropical or subtropical areas. However, climate change and the increased mobility of humans and animals trigger vector migration and the upsurge of parasitic infections in developed countries. Parasitic diseases continue to play a major role in human health, particularly infections caused by protozoans and helminths, such as malaria, trypanosomiasis, leishmaniasis, Chagas disease, schistosomiasis, onchocerciasis, lymphatic filariasis, and helminthiases, which are common in tropical regions and cause millions of morbidities and deaths every year [1,2].

For centuries, nature has been a source of medicines for the treatment of a vast array of diseases, with the first records of the ethnomedicinal uses of plants dating back to 2600 BC in Mesopotamia [3]. The easy access to terrestrial plants helps to explain their popularity as a source of bioactive products and innovative drug leads used in the pharmaceutical industry [4]. The importance of plant-based molecules as antiparasitic agents was reinforced by the Nobel Prize laureates in physiology or medicine in 2015, where Youyou Tu was honored for her discovery of artemisinin, a novel drug for the therapy against malaria, which was derived from *Artemisia annua* L., a plant used in the Chinese traditional medicine [5].

Most research targeting the identification of antiparasitic agents has focused on glycophytes, but salt-tolerant plants (halophytes) have lately aroused the interest of the scientific community due to their multiple medicinal properties, including antiparasitic activities. The harsh habitats where halophytes thrive, such as salt marshes, maritime dunes, and marine cliffs, expose them to extremely variable abiotic conditions, including salinity, light intensity, drought, and temperature [6,7]. This stressful environment contributes to the synthesis and accumulation of bioactive metabolites, including phenolics, alkaloids, and terpenes, conferring to halophyte species important medicinal properties, such as antioxidant, anti-inflammatory, antimicrobial, anti-tumoral, anti-infective, and antiparasitic activities [8,9,10,11,12,13,14,15]. In fact, several halophyte species are used as medicinal (e.g., *Mesembryanthemum edule* L. (syn. *Carpobrotus edulis* L.), and/or dietary plants (e.g., *Chenopodium quinoa* Willd, *Salicornia* spp.), mainly in rural areas where traditional medicine is the only source of health treatments. Moreover, as the problems associated with the salinization of soils and water bodies and the increasing competition for scarce freshwater resources increase [16,17], recruiting wild halophytes with economic potential is one of the suggested strategies to reduce the damage caused by the salinization of soil and water [18]. While most of the crop species used in traditional agriculture are salt sensitive (glycophytes), having a 10% yield decrease as the soil salinity increases over the 4–8 dS/m range, the growth of several halophytes is stimulated within a salinity range of 15–25 dS/m [18]. Halophytes are, therefore, a real strategy as alternative highly salt-tolerant crops that can cope with adverse saline conditions, to be used in the exploitation of degraded agricultural lands with the irrigation with brackish water for sustainable water management and soil conservation when establishing cost-efficient and environmental-friendly agro-ecosystems.

Reports on the traditional medicinal uses of halophytes comprise 43 families and more than 180 species in the Mediterranean, the Arabian Sea, and Syrian regions [12,19,20]. Some of these halophytes, such as *Chenopodium album* and *Artemisia ramosissima* have ethnomedicinal uses for parasitic diseases, including protozoal and helminthic infections [12,18,19], and several scientific reports confirmed their activities by in vitro and in vivo methods and identified their main bioactive constituents. There are several review papers detailing the biological properties of halophytic plant species [12,13,14,15], but information related to the properties of such plants is still dispersed in the literature. Aiming to fulfill this gap, this review provides a comprehensive outline of the ethnomedicinal uses of several species against human parasitic illnesses and the in vitro and in vivo antiprotozoal and anthelmintic properties of such plants, along with the bioactive metabolites that may be responsible for such assets. Finally, a conclusion section is presented, addressing perspectives for the sustainable exploitation of selected species in the context of sustainability and climate change.

## 2. Methodology

This review provides a focused overview of the potential use of halophytes as sources of bioactive molecules and/or natural products to tackle human parasitic diseases. A systematic search was performed to find all English articles with available full text related to the subject from 1993 until December 2022. Searches were performed by consulting several databases, including PubMed, Web of Science, Embase, and Google Scholar (as a search engine). The keywords “ethnomedicinal”, “traditional use”, “halophyte” and/or “salt tolerant plant” were used, alone or in combination with, for example, “antiparasitic”, “antiprotozoal“, and “anthelminthic” and its hyphenated variations. The classification of the plant species as halophytes was confirmed by a search in the eHALOPH database, and/or the description of their occurrence in coastal areas.

## 3. Ethnomedicinal Uses of Halophyte Plants as Antiparasitic Agents

Studies focusing on the ethnomedicinal uses of halophytes are limited [12,19,20,21]. Table 1 summarizes several halophytes species and their ethnopharmacological uses related to the treatment of human parasitic diseases. In brief, species such as *Chenopodium album*, *Artemisia ramosissima, Helichrysum italicum*, *Portulaca oleracea*, and *Limoniastrum monopetalum* are used to treat intestinal helminthic infections in different regions, such as Portugal, Nepal, Pakistan, Libya, Tunisia, the North Sea, India, Spain, and Italy [22,23,24,25,26,27,28,29]. Others, like *Dysphania ambrosioides* and *Portulaca oleracea* are used for their antiprotozoal properties in Albania, Cyprus, Iran, Egypt, and Brazil [28,29]. Several species (e.g., *Limonium vulgare, Portulaca olearacea, Rumex crispus, Elaeagnus ramosíssima, Salsola kali, Dysphania ambrosioides, Chenopodium album*) have been described as anti-diarrheic, which could be related to both helminthic and protozoal parasitic infections [21,22,28,29,30,31,32,33]. Such traditional uses demonstrate the importance and extensive uses of halophyte plants in folk medicine, especially in the Mediterranean region.

Herbal medicine has always had a fundamental role in human welfare, and it is still of utmost importance in many cultures today. In this sense, the ethnopharmacological knowledge of populations can serve as a base to search for novel bioactive compounds. Considering this, scientific research has explored and produced numerous studies about medicinal plants’ biological properties and chemical constituents not only to advance plant drug discovery but also to validate the plant’s traditional uses. However, for halophytes with folk uses, research on their bioactivities and subsequent evidence are still scarce for many of those plants. The next sections review some of the most relevant published work related to antiprotozoal (Section 4) and anthelmintic (Section 5) properties of halophyte species.

## 4. Halophyte Plants as Sources of Antiprotozoal Agents

Aligned to their traditional uses as antiparasitic agents, halophytes have proven by in vitro and in vivo research approaches their potential as sources of molecules with activity towards different protozoa species. Most antiprotozoal studies on natural products focus particularly on neglected tropical diseases (NTDs), a group of twenty infectious illnesses that include, for example, leishmaniasis, human African trypanosomiasis (HAT), Chagas disease, and schistosomiasis. NTDs affect more than 1 billion people worldwide, particularly very poor populations in tropical and subtropical areas in 149 countries [38,39,40]). Leishmaniasis is caused by more than 20 *Leishmania* species, while trypanosomiasis is ascribed to *Trypanosoma*, either the *Trypanosoma brucei* complex (sleeping sickness, human African trypanosomiasis) or *T. cruzi* (Chagas disease, American trypanosomiasis) [40]. Malaria, referred to as a “disease of poverty”, is no longer recognized as an NTD [41] and is caused by protozoa of the genus *Plasmodium*, namely *P. falciparum*, *P. vivax*, *P. malariae*, and *P. ovale*, which are specific for humans [42].

### 4.1. In Vitro Activities and Bioactive Constituents

Most of the reports on the antiparasitic activity of halophyte species include an in vitro screening, followed by the determination of the chemical composition of raw extracts, and less frequently of purified fractions or pure compounds. Sixteen species belonging to 14 different families have been described with in vitro antiprotozoal activity, and those reports are summarized in Table 2.

Essential oils are described as a composite mixture of volatile molecules obtained from aromatic plants, mostly by hydrodistillation, and display highly relevant biological properties, including antiparasitic activities. Essential oils were the main target to evaluate the potential antiprotozoal properties of halophyte species, mostly against *Leishmania* and *Trypanosoma* parasites. For example, the essential oil of flowering aerial parts of *Crithmum maritimum* was highly effective towards *T. brucei* parasites (EC_50_ = 5.0 µg/mL), which was linked to its mono-terpene hydrocarbon content, such as in limonene (EC_50_ = 5.6 µM; Figure 1) and sabinene (EC_50_ = 6.0 µM; Figure 1) [43]. However, they were less effective against *L. infantum* promastigotes (IC_50_ = 122 and 205 µg/mL, respectively) [44]. The antiparasitic features of major compounds, including monoterpene hydrocarbons, sesquiterpene hydrocarbons, oxygen-containing sesquiterpenoids, and diterpenoids, are well described [44]. In turn, the essential oil of leaves and fruits of *Pistacia lentiscus* exerted high inhibitory effects against promastigotes of *Leishmania major, L. tropica*, and *L. infantum*, with IC_50_ values varying between 8 and 26.2 µg/mL [45]. The major volatile components were myrcene and α-pinene in leaves, and α-pinene and limonene, in fruits, all reported with antileishmanial activities [46,47]. In another work, essential oils from leaves of *P. lentiscus* collected from two areas in Tunisia were tested against *L. major* intramacrophage and axenic amastigote forms [48], displaying moderate activities against intracellular amastigote (IC_50_ = 12.5–35.6 µg/mL), and high activity against *L. major* axenic amastigote forms (IC_50_ = 0.5 µg/mL) [48]. The main compounds were identified as pinene, β-myrcene, d-limonene, *O*-cymene, terpinen-4-ol, β-pinene, and α-phellandrene, which may disrupt parasite intracellular metabolic pathways [48].

**Table 2 marinedrugs-21-00066-t002:** In vitro antiprotozoal activity of halophyte species.

Family/Species	Plant Organ	Extract/Fraction/Compound	Chemical Components	Protozoal Species	Results *	References
**Amaranthaceae**						
*Dysphania ambrosioides* (L.) Mosyakin & Clemants (syn. *Chenopodium ambrosioides* L.)	Aerial organs	Essential oil	Terpinolene	*L. amazonensis, L. donovani*	Epimastigotes (IC_50_ = 21.3 µg/mL), and trypomastigotes (IC_50_ = 28.1 µg/mL)	[49]
				*T. cruzi*	Epimastigotes (IC_50_ = 21.3 µg/mL), trypomastigotes (IC_50_ = 28.1 µg/mL), and amastigotes (IC_50_ = 50.2 µg/mL)	[49]
	Aerial parts containing immature seeds	Ethanol ethylacetate extract	Ascaridole [1]; (−)-(2S,4S)-p-mentha-1(7),8-dien-2-hydroperoxide [2]; (−)-(2R,4S)-p-mentha-1(7),8-dien-2-hydroperoxide [3] (−)-(1R,4S)-p-mentha-2,8-dien-1-hydroperoxide [4] (−)-(1S,4S)-p-mentha-2,8-dien-1-hydroperoxide [5].	*T. cruzi* (epimastigotes)	MLC [1] = 23 μM; MLC [2] = 1.2 μM; MLC [3] = 1.6 μM; MLC [4]= 3.1 μM; and MLC [5]= 0.8 μM	[50]
	Leaves	Hydroalchoholic extract	ND	*Giardia lamblia* (trophozoites)	IC_50_ = 198 µg/mL	[33]
	Leaves	70 % Ethanol extract	ND	*Plasmodium falciparum*	IC_50_ = 25.4 μg/mL	[51]
	Leaves	Essential oil	Ascaridole	*Entamoeba histolytica* (trophozoites)	IC_50_ = 700 µg/mL	[52]
**Anacardiaceae**						
*Pistacia lentiscus* L.	Leaves and fruits	Essential oil	Leaves: Myrcene and α-pinene; Fruits: α-pinene and limonene	*Leishmania major*, *L. tropica*, *L. infantum* (clinical isolates)	IC_50_ = 8–26.2 µg/mL	[45]
	Leaves	Essential oil	α-pinene, β-myrcene, D-limonene, o-cymene, terpinen-4- ol, β-pinene, α-phellandrene	*Leishmania major*	Intramacrophage amastigote: IC_50_ = 12.5–35.6 µg/mL; Axenic amastigote: IC_50_ = 0.5–56.1 µg/mL	[48]
**Apiaceae**						
*Crithmum maritimum* L.	Aerial organs	Essential oil	Limonene, γ-terpinene and sabinene	*Trypanossoma brucei*	IC_50_ = 5.0 µg/mL	[43]
	Limonene, sabinene				Limonene: EC_50_ = 5.6 µM Sabinene: EC_50_ = 6.0 µM	
	Aerial organs	Essential oil	α-pinene, *p*-cymene β-phellandrene, Z-β -ocimene, γ-terpinene, thymyl-methyl oxide, dillapiole	*L. infantum* (promastigotes)	IC_50_ = 122 µg/mL	[44]
	Flowers	Decoction	Falcarindiol	*Trypanosoma cruzi*	Extract: EC_50_ = 17.7 µg/mL, SI > 5.65 Fraction: EC_50_ = 0.47 µg/mL, SI = 59.6	[53]
*Eryngium maritimum* L.	Aerial organs	Essential oil	α-pinene, germacrene D, bicyclogermacrene, germacrene, δ-cadinene	*L. infantum* (promastigotes)	IC_50_ = 205 µg/mL	[44]
*Foeniculum vulgare* Mill.	Seeds	Essential oil, *n*-hexane, methanol, and water extracts	E-anethole	*Trichomonas vaginalis*	Methanol and hexane extracts: MLC = 360 µg/mL Essential oil and anethole: MLC = 1600 µg/ml	[54]
	Seeds	Water extract	Hesperidin, ferulic acid, chlorogenic acid	*Blastocystis* spp.	48h: IC_50_ = 224 µg/mL; 72h: IC_50_ = 175 µg/mL	[55]
**Asteraceae**						
*Inula crithmoides* L.	Aerial organs	Dichloromethane extract	Gallic, syringic, salicylic caffeic, coumaric, and rosmarinic acids; epicatechin, epigalocatechin gallate, catechin hydrate, quercetin, and apigenin	*Leishmania infantum*	Intracellular amastigotes: 70% at 125 µg/mL; Promastigotes: 26.5% at 125 µg/mL	[56]
**Caryophyllaceae**						
*Spergularia rubra* (L.) J.Presl & C.Presl and	Aerial organs	Dichloromethane extract	Catechin hydrate	*Leishmania infantum*	Intracellular amastigotes: 25% at 125 µg/mL; promastigotes: 16.7% at 125 µg/mL	[56]
**Cyperaceae**						
*Cyperus rotundus* L.	Tuber of root	Ethyl acetate extract	ND	*Plasmodium falciparum*	Sensitive strain 3D7: IC_50_ = 5.1 µg/mL; resistant strain INDO: IC_50_ = 4 µg/mL	[57]
**Combretaceae**						
*Laguncularia racemosa* (L.) C.F. Gaertn.	Leaves	Chloroform:methanol (1:1) extract	Triterpenoids, phenols	*P. falciparum*	60.1 % at 6.25 μg/mL	[58]
**Fabaceae**						
*Glycyrrhiza glabra* L.	Roots	Water extract	Licochalcone A	* Leishmania major * (promastigotes)	Extract: > 90% death at 1:100 and 1:200 dilutions	[59]
				* L. donovani * (promastigotes)	> 90% death at 1:100 dilution	[59]
		Licochalcone A		* L. major *	Amastigotes: 0 % infection at 5 and 10 μg/mL Promastigotes: 0.4 % at 1:100	[59]
**Juncaceae**						
*Juncus acutus L.*	Roots	Dichloromethane extract and Fraction 8	Phenanthrenes, dihydrophenanthrenes, and benzocoumarins	*Trypanosoma cruzi* (trypomastigotes)	Extract: IC_50_ < 20 µg/mL; Fraction 8: IC_50_ = 4.1 µg/mL, SI: 1.5	[60]
**Nitrariaceae**						
*Peganum harmala* L.	Seeds	Water extract	ND	*L. major* (Promastigotes, amastigotes)	Promastigotes: IC_50_ = 40 µg/mL; Amastigotes: 50% reduction of infection at 10 and 40 µg/mL at 48h	[61]
	Seeds	Hydroalchoholic extract	Harmaline, harmine, and beta-carboline	*L. major* (promastigotes)	IC_50_ = 59.4 µg/mL	[62]
	Seeds	Water extract	ND	*L. donovani* (promastigotes, axenic amastigotes)	Promastigotes: ED_50_ = 458,000 µg/mL at 72 h; Axenic amastigotes: ED_50_ = 6000 µg/mL at 72 h	[63]
	Seeds, Roots	Methanol extract	ND	*L. tropica*	Seeds: IC_50_ = 18.6 µg/mL; Roots: IC_50_ = 16.4 µg/mL	[64]
**Plantaginaceae**						
*Plantago major*	Seeds	80% Ethanol	ND	*P. falciparum*	IC_50_ = 40.0 µg/mL	[65]
**Polygonaceae**						
*Rumex crispus* L.	Leaves, roots	Methanol and ethanol extract	ND	*Trypanosoma brucei brucei*	Etanol root: IC_50_: 9.7 μg/mL	[66]
				*Plasmodium falciparum* 3D7 strain	Methanol leaves: IC_50_ = 15 μg/mL	[66]
**Portulacaceae**						
*Portulaca oleraceae*	Leaves, stems	Essential oils	Phytol, squalene, palmitic acid, ethyllinoleate, ferulic acid, linolenic acid, scopoletin, linoleic acid, rhein, apigenin, bergapten	*L. major* (promastigotes)	Leaves: IC_50_ = 360 µg/mL; Stems: IC_50_ = 680 µg/mL	[67]
** Tetrameristaceae **						
*Pelliciera rhizophorae* Planch. & Triana	Leaves	Methanol:Chloroform (1:1) fraction	α-amyrin, β-amyrine, ursolic acid, oleanolic acid, betulinic acid, brugierol, iso-brugierol, kaempferol, quercetin, and quercetin	*Leishmania donovani*	Oleanolic acid: IC_50_ = 5.3 μM Kaempferol: IC_50_ = 22.9 μM Quercetin:IC_50_ = 3.4 μM	[68]
				*Trypanosoma cruzi*	α-Amyrin: IC_50_ = 19.0 μM	[68]
				*Plasmodium falciparum*	Betulinic acid: IC_50_ = 18.0 μM	[68]

ND: not determined. IC_50_: half-maximal inhibitory concentration; EC_50_: half-maximal effective concentration; LC_50_: median lethal concentration; MLC: minimum lethal concentration; SI = Selectivity index. *: IC_50_ < 15 µg/mL and SI > 3 are considered promising for development as drug leads [69].

Some essential oils from other species have been described with antiparasitic activities, but with higher IC_50_ values, such as those from *P. oleracea* leaves and stems (IC_50_ = 360 and 680 µg/mL) on *L. major* promastigotes [67], or from *F. vulgare* seeds against *T. vaginalis* (MLC = 1600 µg/mL) [54]. The essential oil of *D. ambrosioides* aerial organs have been investigated for its in vitro activity against *L. amazonensis* and *L. donovani*, being highly active towards their epimastigotes (IC_50_ = 21.3 µg/mL) and trypomastigotes (IC_50_ = 28.1 µg/mL), as well as towards *T. cruzi* amastigotes (IC_50_ = 50.2 µg/mL) [49]. Terpinolene was the major active component [49]. In addition, ascaridole, identified as the main component of *D. ambrosioides* leaves’ essential oil, also exhibited in vitro activity against *E. histolytica* parasites, which are responsible for amebiasis, a parasitic disease considered a public health problem in developing countries [52].

Extraction with organic solvents has less expression than the extraction of essential oils, but even so, it also proves to be very effective in the extraction of compounds with antiprotozoal activity from salt-tolerant species. In this context, Oliveira and colleagues [60] performed an in vitro screening of *T. cruzi* trypomastigotes on 94 samples belonging to 31 halophytes species from Southern Portugal. From those, the dichloromethane extract of *Juncus acutus* roots was the most active (IC_50_ < 20 µg/mL), which was further fractionated, affording one active fraction with an IC_50_ of 4.1 µg/mL and selectivity index (SI) of 1.5. The active constituents were identified as phenanthrenes, dihydrophenanthrenes, and benzocoumarins [60]. *C. maritimum* flower decoction also presented anti-*T. cruzi* activity with an EC_50_ value of 17.7 µg/mL and SI of 5.65, and a fraction rich in falcarindiol has an increased activity (EC_50_ = 0.47 µg/mL) and selectivity (SI = 59.6) [53].

The same research group also screened 25 salt-tolerant plant species from southern Portugal for promastigotes and intracellular amastigotes of *L. infantum*, and the highest activity was obtained with the dichloromethane extract of *S. rubra* and *I. crithmoides* aerial organs [56]. The active extracts from *I. crithmoides* were rich in phenolic acids (gallic, syringic, salicylic caffeic, coumaric, and rosmarinic acids) and flavonoids (epicatechin, epigallocatechin gallate, catechin hydrate, quercetin, and apigenin), while catechin hydrate was detected in *S. rubra* [56].

In addition, hexane and methanol extracts of *F. vulgare* seeds were highly active against *T. vaginalis* (MLC = 360 µg/mL) [54], whereas aqueous extracts exhibit anti-*Blastocystis* activity with IC_50_ values ranging between 223.8 and 174.9 µg/mL [55]. The predominant compounds in *F. vulgare* were hesperidin, ferulic, and chlorogenic acid [55]. Several authors reported the in vitro antimalarial properties of *G. glabra*, particularly of root and aerial part extracts [65,70,71]. Furthermore, Licochalcone A (Figure 1) was isolated from its root water extract, with activity against *L. major* amastigotes (0% infected cells at 5 and 10 µg/mL) and promastigotes (0.4% infected cells at 1:100). Moreover, major components of a methanol:chloroform fraction obtained from the leaves of the tea mangrove *Pelliciera rhizophorae*, showed high antiprotozoal activity against *L. donovani* (Oleanolic acid: IC_50_ = 5.3 μM; Kaempferol: IC_50_ = 22.9 μM; Quercetin: IC_50_ = 3.4 μM), *T. cruzi* (α-amyrin: IC_50_ = 19.0 μM), as well as *P. falciparum* (Betulinic acid: IC_50_ = 18.0 μM) (Figure 1) [68].

Other authors reported the in vitro anti-*Plasmodium* efficacy of ethanolic extracts of *Plantago major* seeds (*P. falciparum*: IC_50_ = 40.0 µg/mL) [65] and *C. rotundus* tuber root ethyl acetate extract (*P. falciparum* IC_50_ = 5.1 µg/mL and 4 µg/mL for sensitive and resistant strains, respectively) [57]. *Peganum harmala* seeds and aerial parts extracts have been extensively studied for their antileishmanial properties against *L. major*, *L. donovani*, and *L. tropica* as sustained by different authors [61,62,63,64].

Overall, some authors defined that extracts with IC_50_ values below 15 µg/mL and selectivity above 3 can be considered promising for further development as drug leads [69]. Following this guideline, the fraction of flower decoction of *C. maritimum* can be considered the most promising sample with anti-*T. cruzi* activity by coupling both criteria (EC_50_ = 0.47 µg/mL; SI = 59.6) [53].

### 4.2. In Vivo Studies

Only a few authors evaluated the in vivo antiprotozoal potential of halophytes, including only three species and using mainly rodents as models. These reports are summarized in Table 3.

The most studied species was *D. ambrosioides*, investigated by five different authors. For instance, essential oil from its aerial parts was more effective against experimental cutaneous leishmaniasis by *L. amazonensis* in BALB/c mice than its pure main components, namely ascaridole, carvacol, and caryophyllene oxide [72]. In turn, hydroalcoholic extracts of its leaves displayed in vivo antimalarial properties against BALB/c mice infected with *P. berghei* intraperitoneally [51], and in vivo effects on C3H/HePas mice infected with *L. amazonensis* promastigotes [73]. Moreover, ascaridole was the main component in *D. ambrosioides* leaves’ essential oil and exhibited in vitro and in vivo activity against *E. histolytica* parasites [52]. Besides, other authors reported the anti-*Plasmodium* efficacy of ethanolic extracts of *A. officinalis* flowers (*P. falciparum* IC_50_ = 62.7 µg/mL; *P. berghei* suppression of parasitemia in vivo [400 mg/kg] = 62.86 %), and of *P. major* seeds (*P. falciparum*: IC_50_ = 40.0 µg/mL; *P. berghei* suppression of parasitemia in vivo [400 mg/kg] = 22.5%) [65].

**Table 3 marinedrugs-21-00066-t003:** In vivo antiprotozoal activity of halophyte species.

Family/Species	Plant Organ	Extract/Fraction/Compound	Chemical Components	Assay	Results	References
** Amaranthaceae **						
*Dysphania ambrosioides* (L.) Mosyakin & Clemants (syn. *Chenopodium ambrosioides* L.)	Aerial organs	Essential oils	Ascaridole, carvacrol, caryophyllene oxide	Cutaneous leishmaniasis-*L. amazonensis* in BALB/c mice	Prevented lesion development compared with untreated animals	[72]
		Mix of ascaridole, carvacrol, caryophyllene oxide		Cutaneous leishmaniasis-*L. amazonensis* in BALB/c mice	Cause death of animals after 3 days of treatment	[72]
	Leaves	Essential oil	Ascaridole	*Entamoeba histolytica* HM-1 in IMSS strain Golden hamsters infected with trophozoites	8 mg/kg and 80 mg/kg reverted the infection	[52]
	Leaves	70% Ethanol	ND	BALB/c mice infected with *P. berghei*	Increased survival and decreased parasitaemia	[51]
	Leaves	70% Ethanol	ND	C3H/HePas mice infected with *Leishmania amazonensis* promastigotes	Reduced nitric oxide production and the parasite load	[73]
**Malvaceae**						
*Althaea officinalis* L.	Flowers	80% Ethanol	ND	*P. berghei* infected female Swiss albino mice	Suppression of parasitemia = 62.86 %, at a dose of 400 mg/kg	[65]
**Plantaginaceae**						
*Plantago major* L.	Seeds	80% Ethanol	ND	*P. berghei* infected female Swiss albino mice	Suppression of parasitemia = 22.46 %, at a dose of 400 mg/kg	[65]

ND: not determined.

## 5. Halophyte Plants as Sources of Anthelmintic Agents

Supported by their traditional uses, some halophyte species have also demonstrated in vitro and in vivo potential as sources of molecules with anthelmintic activity. Most anthelmintic studies on natural products focus mainly on the model nematode *Caenorhabditis elegansi*, or gastrointestinal nematodes (GINs), namely *Haemonchus contortus* and *Trichostrongylus colubriformis*, a leading cause of production loss in agricultural animal systems worldwide [74,75]. Since different *Trichostrongylus* species, as for example, *T. colubriformis*, can infect humans in different areas of the world (e.g., Iran, Laos, Australia) [76], that species was included in this review.

### 5.1. In Vitro Activities and Bioactive Constituents

Most of the reports on the anthelmintic activity of halophytes were performed only in vitro, and only less than half included a chemical characterization of its major constituents. Twelve species from 9 families have been described with in vitro anthelmintic properties, and those reports are included in Table 4.

Oliveira and colleagues [77] have screened 80% acetone extracts from 8 halophyte species for their anthelmintic capacity using the Larval Ensheathment Inhibition Assay (LEIA) and Egg Hatching Inhibition Assay (EHIA) for *T. colubriformis*. *P. lentiscus*, *L. monopetalum*, *C. mariscus,* and *H. italicum picardi* were the most active in both GINs and life stages of the nematodes [77]. In particular, *C. mariscus* aerial parts collected during summer were more active against *T. colubriformis* (EC_50_ = 77.8 µg/mL) [78]. Moreover, inflorescences presented the highest anthelmintic activity against *T. colubriformis* [EC_50_ (LEIA) = 78.6 µg/mL; IC_50_ (EHIA) = 848.2 µg/mL]. Flavan-3-ols, proanthocyanidins, luteolin, and glycosylated flavonoids are the main active components [78]. Some authors have also tested some halophyte extracts against *S. mansoni,* which is a helminth species and the causal agent of schistosomiasis in humans. Methanol extracts of *C. ambrosioides* reduced *S. mansoni* cercariae infectivity [79], as well as essential oils from *F. vulgare* that showed moderate in vitro activity against *S. mansoni* worms, but with more remarkable effects in egg development, possibly attributable to the two main constituents detected in the essential oil, I-anethole and limonene [80]. A compound isolated from *G. inflata*, licochalcone A (Figure 2), presented an LC_50_ of 9 μM towards *S. mansoni* female and male adult worms [81].

Besides, other authors have focused on other less frequently tested parasite species. For instance, the roots of *G. glabra* contain glycyrrhizic acid (Figure 2) that is very active in vitro against *B. malayi* microfilarae (IC_50_ = 1.20 μM), one of the most important causative agents of human lymphatic filariasis [82].

**Table 4 marinedrugs-21-00066-t004:** In vitro anthelmintic activity of halophyte species.

Family/Species	Plant Organ	Extract/Fraction/Compound	Chemical Components	Assay	Results	References
**Apiaceae**						
*Foeniculum vulgare* Mill.	Fresh leaves	Essential oil	I-anethole and limonene	*Schistosoma mansoni* adult worms (pairs) and eggs	50% activity at 100,000 µg/mL (24 and 120 h)	[80]
**Asteraceae**						
*Helichrysum italicum* (Roth) G. Don subsp. *picardi* (Boiss. & Reut.) Franco	Aerial parts	80% acetone extract	Caffeoylquinic and dicaffeoylquinic acids and quercetin glycosides	*Trichostrongylus colubriformis*	IC_50_ (LEIA) = 132 µg/mL; IC_50_ (EHIA) = 3707 µg/mL	[77]
*Inula crithmoides* L.	Aerial parts	80% acetone extract	ND	*Trichostrongylus colubriformis*	IC_50_ (LEIA) = 1031 µg/mL	[77]
**Cyperaceae**						
*Cladium mariscus* L. Pohl	Aerial parts	80% acetone extract	Proanthocyanins, phenolic acids, and luteolin	*Trichostrongylus colubriformis*	IC_50_ (LEIA) = 77.8 µg/mL; IC_50_ (EHIA) = 2575 µg/mL	[77]
	Aerial parts, leaves, and inflorescences collected during spring, summer, autumn, and winter	80% acetone extract	Flavan-3-ols, proanthocyanidins, luteolin, and glycosylated flavonoids	*Trichostrongylus colubriformis*	Summer: EC_50_ (LEIA) = 77.8 µg/mL; Spring: IC_50_ (EHIA) = 2275 µg/mL; Leaves: EC_50_ (LEIA) = 81.1 µg/mL; IC_50_ (EHIA) = 2289 µg/mL; Inflorescences: EC_50_ (LEIA) = 78.6 µg/mL; IC_50_ (EHIA) = 848 µg/mL	[78]
** Convolvulaceae **						
*Calystegia soldanela* (L.) R. Br.	Aerial parts	80% acetone extract	ND	*Trichostrongylus colubriformis*	IC_50_ (LEIA) = 2711 µg/mL	[77]
** Fabaceae **						
*Glycyrrhiza glabra* L.	Roots	Glycyrrhizic acid		*Brugia malayi* microfilarae in vitro	IC_50_ = 1.20 μM	[82]
*Glycyrrhiza inflata* Batalin	ND	Licochalcone A		*S. mansoni* (female and male adult worms)	LC_50_ = 9 μM	[81]
* Medicago marina * L.	Aerial parts	80% acetone extract	ND	*Trichostrongylus colubriformis*	IC_50_ (LEIA) = 211 µg/mL	[77]
** Plantaginaceae **						
* Plantago coronopus * L.	Aerial parts	80% acetone extract	ND	*Trichostrongylus colubriformis*	IC_50_ (LEIA) = 212 µg/mL	[77]
** Plumbaginaceae **						
* Limoniuastrum monopetalum * (L.) Boiss.	Aerial parts	80% acetone extract	Sulphated and/or methylated flavonoids	*Trichostrongylus colubriformis*	IC_50_ (LEIA) = 1024 µg/mL; IC_50_ (EHIA) = 2102 µg/mL	[77]
**Poaceae**						
*Cynodon dactylon* (L.) Pers.	ND	Methanol extract	ND	*Hymenolepis* *diminuta*	40,000 µg/mL: paralysis and mortality at 4.12 h and 5.16 h, respectively	[83]
** Rubiaceae **						
*Crucianella marítima *L.	Aerial parts	80% acetone extract	ND	*Trichostrongylus colubriformis*	IC_50_ (LEIA) = 1024 µg/mL	[83]

ND: not determined.

### 5.2. In Vivo Studies

There is a reduced number of studies focusing on in vivo anthelmintic properties of halophytic species. In fact, to our best knowledge, only three species were tested, mostly using rodents and sheep as models (Table 5).

*Pistacia lentiscus* is the most studied halophyte species due to its high level of tannins, known to exhibit high anthelmintic activity, as confirmed by in vivo studies against *T. colubriformis*. Infected goats fed with aerials parts of *P. lentiscus* presented a 16% reduction in fecal oocyst counts, whereas, in lambs, the reduction ranged between 55.2 and 61.3% [84,85]. In turn, *S. mansoni* infected mice treated orally with a methanol extract of *C. ambrosioides* (1250 mg/kg/day) exhibited a 53.7% total worm burden decrease and a 60.3% ova/g tissue in liver reduction, 9 weeks post-infection [86]. *Hymenolepis diminuta* infected Wistar rats were treated with *C. dactylon* methanol extract (800 mg/kg) enabling a reduction of 77.6% in eggs and 79% in adult worms counts, respectively [83].

**Table 5 marinedrugs-21-00066-t005:** In vivo anthelmintic activity of halophyte species.

Family/Species	Plant Organ	Extract/Fraction/Compound	Chemical Components	Assay	Results	References
** Amaranthaceae **						
*Dysphania ambrosioides* (L.) Mosyakin & Clemants (syn. *Chenopodium ambrosioides* L.)	ND	Methanol	ND	*S. mansoni* infected mice	1250 mg/kg/day exhibited a 53.7% total worm burden decrease and a 60.3% ova/g tissue in liver reduction	[79,86]
**Anacardiaceae**						
*Pistacia lentiscus* L.	Aerial parts	ND	Tannins	*Teladorsagia circumcincta, Trichostrongylus colubriformis*, and *Chabertia ovina* infected goats	Reduced fecal oocyst counts in approx. 16%	[85]
		ND	Tannins	*T. colubriformis* infected lambs	Reduction of 55.2–61.3% on faecal egg counts	[84]
**Poaceae**						
*Cynodon dactylon* (L.) Pers.	ND	Methanol extract	ND	*Hymenolepis* *diminuta* infected Wistar rats	800 mg/kg: 77.6 and 79% reduction in egg and worms’ reduction, respectively	[83]

ND: Not determined.

## 6. Conclusions

The importance of halophytes as sources of food and medicinal commodities is rising. Their role is of particular importance in the context of soil and water salinization since they can be commercially cultivated in underutilized saline areas and under sustainable saline agricultural systems. This review collected for the first time data on halophytes’ ethnomedical properties towards human parasitic infections, as well as the existing scientific reports on its in vitro and in vivo experiments on antiprotozoal and anthelmintic properties. Sixteen species from 14 different families have reported antiprotozoal properties, with most studies focusing on different strains of *Leishmania*, *Trypanosoma* and *Plasmodium* parasites. In this context, eight pure compounds occurring in halophyte species were identified with antiprotozoal properties, namely limonene and sabinene against *T. brucei,* licochalcone A against *L. major*, oleanolic acid, kaempferol and quercetin anti-*L. donovani,* and betulinic acid and α-amyrin towards *T. cruzi* and *P. falciparum*, respectively.

In turn, 15 species from 12 families were studied for their anthelmintic capacity, mainly against *Trichostrongylus* gastrointestinal nematodes. Essential oils were the most represented type of extract, followed by extracts made with different combinations of water and organic solvents. From the in vivo studies, the most promising species is *D. ambrosioides*, including essential oil from aerial organs towards cutaneous leishmaniasis and hydroalcoholic extracts of its leaves, with antimalarial properties. Ascaridole, the major metabolite detected in *D. ambrosioides* leaves essential oil, exhibited in vitro and in vivo activity against *E. histolytica* parasites, responsible for amebiasis. Regarding anthelmintic properties, the most promising species is *P. lentiscus*, as confirmed by in vivo studies of *T. circumcincta, and T. colubriformis* nematodes. Furthermore, a few compounds were isolated from halophyte species with anthelmintic properties, such as licochalcone A from *G. inflata* with anti-*S. mansoni* activity, and glycyrrhizic acid obtained from *G. glabra* roots with in vitro activity against *B. malayi* microfilariae. Therefore, halophytes have shown to be promising sources of compounds with potential application as antiparasitic commodities for both humans and animals, including neglected tropical diseases related to protozoan, and gastrointestinal nematodes infections. The main gap in the search for halophytic products with antiparasitic activity is the isolation and identification of novel molecules. Thus, efforts must be made in this regard.

## Figures and Tables

**Figure 1 marinedrugs-21-00066-f001:**
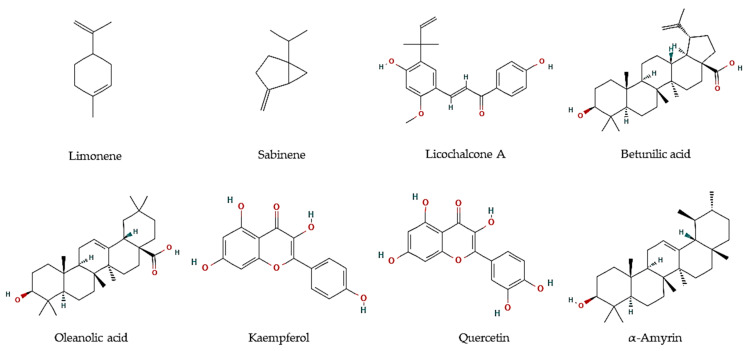
Chemical structures of pure compounds reported with antiprotozoal properties.

**Figure 2 marinedrugs-21-00066-f002:**
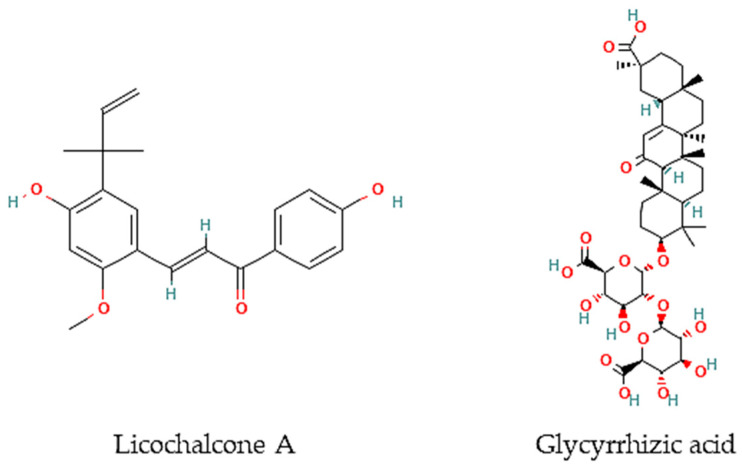
Chemical structures of pure compounds reported with anthelmintic properties.

**Table 1 marinedrugs-21-00066-t001:** Examples of halophytes used in ethnomedicine towards parasitic diseases.

Plant Family/Species	Medicinal Use	Plant Organs/Administration	Country/Region	References
**Amaranthaceae**				
*Chenopodium album* L. (syn. *Chenopodium centrorubrum* Nakai; *Chenopodium virgatum* Thunb.; and *Chenopodium iljinii* Golosk)	Intestinal worms	Shoots, juice	Nepal	[28]
	Anthelminthic	Whole plant	Pakistan	
	Antidiarrhea	Whole plant, decoction	Pakistan	
	Anthelmintic	Whole plant	Pakistan	
	Intestinal worms	Leaves, cooking	Pakistan	
*Dysphania ambrosioides* (L.) Mosyakin & Clemants (syn. *Chenopodium ambrosioides* L.)	Antidiarrhea, cutaneous leishmaniasis	Leaves	Brazil	[33,34]
*Salsola kali* L.	Antidiarrhea	ND	Cyprus; Tunisia North Sea	[21,22,31]
**Apiaceae**				
*Artemisia ramosissima* L. ssp *ramosíssima* Arcangeli	Anthelmintic	Stems and leaves infusions and decoctions	Portugal	[27]
	Anthelmintic, insecticide	ND	North Sea; India	[22,26]
*Helichrysum italicum* (Roth) G.Don	Anthelmintic	Leaves and flowers infusions and decoctions, essential oils	Italy, Spain, Portugal	[23,25]
**Elaeagnaceae**				
*Elaeagnus ramosíssima* L.	Antidiarrhea	Fruits	Iran	[32]
**Fabaceae**				
*Glycyrrhiza glabra* L.	Insecticide			[35,36]
**Polygonaceae**				
*Rumex crispus* L.	Antidiarrhea	Seeds infusions	Portugal	[30]
**Portulacaceae**				
*Portulaca olearacea* L.	Antiparasitic	Roots, stems, leaves Boiling leaves’ vapor	Albania, Cyprus Iran, Egypt	[31,32,37]
	Antidiarrhea	Seeds	Afghanistan	[29]
	Vermifuge	Aerial parts	Pakistan	
	Antidiarrhoea	Leaves	Turkey	
	Vermicide		Libya	
**Plumbaginaceae**				
*Limoniastrum monopetalum* (L.) Boiss.	Antiparasites that cause painful and bloody diarrhea	Leaves and galls infusions	Tunisia	[24]
*Limonium vulgare* Mill.	Anti-diarrhea	ND	North Sea	[22]

ND: Not determined.

## Data Availability

Data is contained within the article.

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
