# Peer review of "Salt-Tolerant Plants as Sources of Antiparasitic Agents for Human Use: A Comprehensive Review"

_marinedrugs, 2023, doi:10.3390/md21020066_

Round 1

Reviewer 1 Report

This article is  Accepted  in present form.

Author Response

We are grateful to the reviewer for this comment. 

Reviewer 2 Report

1. For the reported EC50 values, the authors should use either units of µg/mL or µM, but not both. The former is likely more convenient since many of these are for compound mixtures.  In addition, the authors report EC values with units of mg/mL and ppm which should not be used.

2. A consistent number of significant figures should be used for the EC50 values, probably no more than two.

3. The authors should report and discuss cytotoxicity data whenever it is available so the reader can get some idea of antiparasitic selectivity.

Author Response

REVIEWER 2

  1. For the reported EC50 values, the authors should use either units of µg/mL or µM, but not both. The former is likely more convenient since many of these are for compound mixtures.  In addition, the authors report EC values with units of mg/mL and ppm which should not be used.

 Ans.: We consider more correct to keep the original units, µg/mL extracts or fractions, and µM for pure compounds. However, the values with units of mg/mL were changed to µg/mL.

  1. A consistent number of significant figures should be used for the EC50 values, probably no more than two.

 Ans.: The number of significant figures was uniformized.

  1. The authors should report and discuss cytotoxicity data whenever it is available so the reader can get some idea of antiparasitic selectivity.

 Ans.: When available the selectivity index was included in the text.

Reviewer 3 Report

The authors provide a comprehensive review of the potential medicinal use of compounds derived from salt-tolerant halophytes to treat human parasitic diseases. The background section provides adequate information. The methods section seems to be too short; it can be further developed. The results and discussion are well presented, both in the tables and figures.

English needs some corrections.

MAJOR COMMENTS:

Methods, Lines 82-91: Please provide more information on how the articles were selected, for example, from what year to what year. If necessary, the authors may want to consult the complete PRISMA checklist (http://www.prisma-statement.org/) and consider providing the flow diagram to show how the articles were selected for or excluded from this review paper.

Table 2: Under the column “protozoal species” the authors placed Anopheles stephensi larvae and Aedes aegypti larvae. These are not protozoans, but insects. I suggest removing the insects from this paper to keep the review focused on protozoa and helminths. The related statements on these malaria vectors in the main text (lines 167-171) can also be deleted.

MINOR COMMENTS:

Line 2, article title, “antiparasitic commodities”: I am not sure if this expression is appropriate. The more commonly used expressions would be antiparasitic agents or drugs or candidate drugs.

Lines 14-15, “efforts targeting the identification of antiparasitic drugs from plants has…”: efforts…have

Line 28: A period after “areas” then a new sentence from “However,”

Lines 30-35, “Though, parasitic diseases…”: Please write a complete sentence. “Though,” can be deleted to make it into a complete sentence.

Line 47: including antiparasitic properties/actions/activities

Line 53: antioxidant, anti-inflammatory, … and antiparasitic activities [8-15]

Line 70: some of these halophytes…ethnomedicinal uses for parasitic disease

Line 90: The classification…was confirmed

Line 97: intestinal helminthic infection

Line 98: “North Sea” is not a country. The authors can say country or region, as they have done in Table 1.

Line 105: folk medicine, especially in the Mediterranean region

Line 112-113: research… and subsequent evidence are still scarce

Line 114, “The next sections review…antiparasitic properties of halophyte species”: This sentence is in section 3. “Ethnomedicinal uses of halophyte plants as antiparasitic agents.” The next section that the authors are referring to is section 4. “Halophyte plants as sources of antiprotozoal agents.” Instead of “The next sections review…antiparasitic properties of halophyte species” I think that the authors mean “The next sections review…antiprotozoal (not “antiparasitic”) properties of halophyte species.”

Line 126: Leishmaniasis is caused

Lines 127-128: either the Trypanosoma brucei complex or T. cruzi

Lines 130-131, “Plasmodium…which are specific for humans [42]”: REF 42 is about Trichostrongylus colubriformis. Please re-check whether this reference supports the statement in lines 130-131.

Line 140: including antiparasitic properties/action/activities

Lines 142-145 “the essential oil…was highly effective towards T. brucei parasites…and sabinene highly active towards T. brucei parasites”: The phrases “highly effective towards T. brucei parasites” and “highly active towards T. brucei parasites” within the same sentence are redundant. The second phrase “highly active towards T. brucei parasites” can be deleted.

Minor comments on Table 2:

-Please add a legend and explain what the abbreviations stand for: IC50, 50% inhibitory concentration; EC50; MLC; LC50.

-In addition, the unit of measure for Aedes aegypti (LC50 expressed as “µL/L”) is not clear.

-For the row Cyperaceae, Plasmodium falciparum first, then under it, P. falciparum.

Line 194, “were highly active samples towards T. vaginalis”: The word “samples” can be deleted.

Line 207: Other authors reported

Line 207, “P. major seeds”: Please provide the genus name of P. major in the text, even if it can be found in the table (Plantago major).

Lines 227-228, “E. histolytica parasites, responsible for amebiasis, a parasitic disease considered a public health problem in developing countries”: The authors already mentioned in lines 176-178: “E. histolytica parasites, which are responsible for amebiasis, a parasitic disease considered a public health problem in developing countries.” In lines 227-228, the authors can simply say: “…against E. histolytica parasites [52].”

Line 228, anti-Plasmodium: “Plasmodium” in italic

Line 230-231, “P. berghei suppression of parasitemia”: It would be helpful to know in the main text the concentrations/dose of the extracts of A. officinalis flowers and P. major seeds given to attain these results even though the information is given in Table 3.

Lines 240-241, “Trichostrongylus species” and “in different areas”: “species” and “in” not in italics

Line 253, “against the species S. mansoni”: The authors can say more simply – against S. mansoni

Lines 258-260, “Whereas a compound…”: Please delete “whereas” so that this sentence becomes a complete sentence.

Line 279, “1250 mg/Kg/day”: 1250 mg/kg/day; see also in Table 5

Line 280, H. diminuta infected Wistar rats”: Hymenolepis diminuta

Line 313, “Though, the main gap…”: The word “Though” can be deleted so that the sentence becomes a complete sentence. A period can be placed after “novel molecules.” Then start a new sentence “Thus, efforts must be made in this regard.”

Tables 2-5: In each of these tables, it would be helpful if the authors can add (in the table legend) the concentration or dose that is considered to be promising for further development of the candidate drugs.

Please use the same format for all references. See REF 2, 6, 15, 17, 28, 35, 46, 52, 54, 56, 57, 61, 63, 68, 72, and 80. The first letters of each word in the article title are in capital letter, unlike in other references.

REF 5, REF 27: Please provide the author (or write “anonymous”) and follow the format used by the journal for web links.

REF 14 is incomplete: Ocean Coast Manag 2022, 225, 106228.

REF 53 is incomplete: Plants 2021, 10, 2235.

REF 81: journal name – Please italicize “Pestic.”

REF 89: Please delete “89” and “90”

Author Response

REVIEWER 3

MAJOR COMMENTS:

 Methods, Lines 82-91: Please provide more information on how the articles were selected, for example, from what year to what year. If necessary, the authors may want to consult the complete PRISMA checklist (http://www.prisma-statement.org/) and consider providing the flow diagram to show how the articles were selected for or excluded from this review paper.

Ans.: Details regarding the interval of years of selected papers were included in the methodology.

 Table 2: Under the column “protozoal species” the authors placed Anopheles stephensi larvae and Aedes aegypti larvae. These are not protozoans, but insects. I suggest removing the insects from this paper to keep the review focused on protozoa and helminths. The related statements on these malaria vectors in the main text (lines 167-171) can also be deleted.

 Ans.: The referred insect species were removed from the tables and related statements.

 MINOR COMMENTS:

 Line 2, article title, “antiparasitic commodities”: I am not sure if this expression is appropriate. The more commonly used expressions would be antiparasitic agents or drugs or candidate drugs.

Ans.: The title was changed to “…antiparasitic agents” as suggested.

 Lines 14-15, “efforts targeting the identification of antiparasitic drugs from plants has…”: efforts have

Ans.: This was corrected.

 Line 28: A period after “areas” then a new sentence from “However,”

Ans.: This was corrected.

 Lines 30-35, “Though, parasitic diseases…”: Please write a complete sentence. “Though,” can be deleted to make it into a complete sentence.

Ans.: This was corrected.

 Line 47: including antiparasitic properties/actions/activities

Ans.: This was corrected.

 Line 53: antioxidant, anti-inflammatory, … and antiparasitic activities [8-15]

Ans.: This was corrected.

 Line 70: some of these halophytes…ethnomedicinal uses for parasitic disease

Ans.: This was corrected.

 Line 90: The classification…was confirmed

Ans.: This was corrected.

 Line 97: intestinal helminthic infection

 Ans.: This was corrected.

Line 98: “North Sea” is not a country. The authors can say country or region, as they have done in Table 1.

Ans.: This was corrected.

 Line 105: folk medicine, especially in the Mediterranean region

Ans.: This was corrected.

 Line 112-113: research… and subsequent evidence are still scarce

Ans.: This was corrected.

 Line 114, “The next sections review…antiparasitic properties of halophyte species”: This sentence is in section 3. “Ethnomedicinal uses of halophyte plants as antiparasitic agents.” The next section that the authors are referring to is section 4. “Halophyte plants as sources of antiprotozoal agents.” Instead of “The next sections review…antiparasitic properties of halophyte species” I think that the authors mean “The next sections review…antiprotozoal (not “antiparasitic”) properties of halophyte species.”

Ans.: This was corrected.

 Line 126: Leishmaniasis is caused

Ans.: This was corrected.

 Lines 127-128: either the Trypanosoma brucei complex or T. cruzi             

Ans.: This was corrected.

 Lines 130-131, “Plasmodium…which are specific for humans [42]”: REF 42 is about Trichostrongylus colubriformis. Please re-check whether this reference supports the statement in lines 130-131.

Ans.: By mistake, the mentioned citation was referring to the wrong reference. It was altered to the right one.

 Line 140: including antiparasitic properties/action/activities

Ans.: This was corrected.

 Lines 142-145 “the essential oil…was highly effective towards T. brucei parasites…and sabinene highly active towards T. brucei parasites”: The phrases “highly effective towards T. brucei parasites” and “highly active towards T. brucei parasites” within the same sentence are redundant. The second phrase “highly active towards T. brucei parasites” can be deleted.

 Ans.: This was corrected.

Minor comments on Table 2:

-Please add a legend and explain what the abbreviations stand for: IC50, 50% inhibitory concentration; EC50; MLC; LC50.

Ans.: This information was included in tables legend.

-In addition, the unit of measure for Aedes aegypti (LC50 expressed as “µL/L”) is not clear.

Ans.: According to your previous comment, information regarding insects was removed from the table.

-For the row Cyperaceae, Plasmodium falciparum first, then under it, P. falciparum.

 Ans.: This was corrected.

Line 194, “were highly active samples towards T. vaginalis”: The word “samples” can be deleted.

 Ans.: This was corrected.

Line 207: Other authors reported

 Ans.: This was corrected.

Line 207, “P. major seeds”: Please provide the genus name of P. major in the text, even if it can be found in the table (Plantago major).

 Ans.: This was corrected.

Lines 227-228, “E. histolytica parasites, responsible for amebiasis, a parasitic disease considered a public health problem in developing countries”: The authors already mentioned in lines 176-178: “E. histolytica parasites, which are responsible for amebiasis, a parasitic disease considered a public health problem in developing countries.” In lines 227-228, the authors can simply say: “…against E. histolytica parasites [52].”

 Ans.: This was corrected.

Line 228, anti-Plasmodium: “Plasmodium” in italic

 Ans.: This was corrected.

Line 230-231, “P. berghei suppression of parasitemia”: It would be helpful to know in the main text the concentrations/dose of the extracts of A. officinalis flowers and P. major seeds given to attain these results even though the information is given in Table 3.

 Ans.: This was corrected.

Lines 240-241, “Trichostrongylus species” and “in different areas”: “species” and “in” not in italics

Ans.: This was corrected. 

Line 253, “against the species S. mansoni”: The authors can say more simply – against S. mansoni

 Ans.: This was corrected.

Lines 258-260, “Whereas a compound…”: Please delete “whereas” so that this sentence becomes a complete sentence.

 Ans.: This was corrected.

Line 279, “1250 mg/Kg/day”: 1250 mg/kg/day; see also in Table 5

 Ans.: This was corrected.

Line 280, H. diminuta infected Wistar rats”: Hymenolepis diminuta

 Ans.: This was corrected.

Line 313, “Though, the main gap…”: The word “Though” can be deleted so that the sentence becomes a complete sentence. A period can be placed after “novel molecules.” Then start a new sentence “Thus, efforts must be made in this regard.”

 Ans.: This was corrected.

Tables 2-5: In each of these tables, it would be helpful if the authors can add (in the table legend) the concentration or dose that is considered to be promising for further development of the candidate drugs.

 Ans.: This information was included in table legend, and a sentence referring to the most promising samples identified in the review was included in the manuscript.

Please use the same format for all references. See REF 2, 6, 15, 17, 28, 35, 46, 52, 54, 56, 57, 61, 63, 68, 72, and 80. The first letters of each word in the article title are in capital letter, unlike in other references.

 Ans.: This was corrected.

REF 5, REF 27: Please provide the author (or write “anonymous”) and follow the format used by the journal for web links.

 Ans.: The references already follow the journal format for weblinks: “Title of Site. Available online: URL (accessed on Day Month Year)”.

REF 14 is incomplete: Ocean Coast Manag 2022, 225, 106228.

 Ans.: This reference was corrected.

REF 53 is incomplete: Plants 2021, 10, 2235.

 Ans.: This reference was corrected.

REF 81: journal name – Please italicize “Pestic.”

 Ans.: This reference was corrected.

REF 89: Please delete “89” and “90”

 Ans.: This was removed.